# Germ-Free Conditions Modulate Host Purine Metabolism, Exacerbating Adenine-Induced Kidney Damage

**DOI:** 10.3390/toxins12090547

**Published:** 2020-08-26

**Authors:** Eikan Mishima, Mariko Ichijo, Takeshi Kawabe, Koichi Kikuchi, Yukako Akiyama, Takafumi Toyohara, Takehiro Suzuki, Chitose Suzuki, Atsuko Asao, Naoto Ishii, Shinji Fukuda, Takaaki Abe

**Affiliations:** 1Division of Nephrology, Endocrinology, and Vascular Medicine, Tohoku University Graduate School of Medicine, Sendai 980-8574, Japan; eikan@med.tohoku.ac.jp (E.M.); mariko.ichi@med.tohoku.ac.jp (M.I.); koichikikuchi@med.tohoku.ac.jp (K.K.); yukako.akiyama@med.tohoku.ac.jp (Y.A.); toyohara@med.tohoku.ac.jp (T.T.); suzuki2i@med.tohoku.ac.jp (T.S.); chitose@med.tohoku.ac.jp (C.S.); 2Department of Microbiology and Immunology, Tohoku University Graduate School of Medicine, Sendai 980-8574, Japan; kawabet@med.tohoku.ac.jp (T.K.); aasao@med.tohoku.ac.jp (A.A.); ishiin@med.tohoku.ac.jp (N.I.); 3Institute for Advanced Biosciences, Keio University, Tsuruoka 997-0052, Japan; sfukuda@sfc.keio.ac.jp; 4Intestinal Microbiota Project, Kanagawa Institute of Industrial Science and Technology, Kawasaki 210-0821, Japan; 5Transborder Medical Research Center, University of Tsukuba, Tsukuba 305-8575, Japan; 6Division of Medical Science, Tohoku University Graduate School of Biomedical Engineering, Sendai 980-8574, Japan; 7Department of Clinical Biology and Hormonal Regulation, Tohoku University Graduate School of Medicine, Sendai 980-8574, Japan

**Keywords:** microbiota, uremic toxins, xanthine dehydrogenase, xanthine oxidase, uric acids, chronic kidney disease, gut-kidney axis, IL-17, Th17

## Abstract

Alterations in microbiota are known to affect kidney disease conditions. We have previously shown that germ-free conditions exacerbated adenine-induced kidney damage in mice; however, the mechanism by which this occurs has not been elucidated. To explore this mechanism, we examined the influence of germ-free conditions on purine metabolism and renal immune responses involved in the kidney damage. Germ-free mice showed higher expression levels of purine-metabolizing enzymes such as xanthine dehydrogenase, which converts adenine to a nephrotoxic byproduct 2,8-dihydroxyadenine (2,8-DHA). The germ-free mice also showed increased urinary excretion of allantoin, indicating enhanced purine metabolism. Metabolome analysis demonstrated marked differences in the purine metabolite levels in the feces of germ-free mice and mice with microbiota. Furthermore, unlike the germ-free condition, antibiotic treatment did not increase the expression of purine-metabolizing enzymes or exacerbate adenine-induced kidney damage. Considering renal immune responses, the germ-free mice displayed an absence of renal IL-17A expression. However, the adenine-induced kidney damage in wild-type mice was comparable to that in IL-17A-deficient mice, suggesting that IL-17A does not play a major role in the disease condition. Our results suggest that the enhanced host purine metabolism in the germ-free mice potentially promotes the conversion of the administered adenine into 2,8-DHA, resulting in exacerbated kidney damage. This further suggests a role of the microbiota in regulating host purine metabolism.

## 1. Introduction

Recent evidence has highlighted the importance of gut microbiota in the pathophysiology of kidney diseases and proposed a relationship between the gut and kidney, named the gut-kidney axis [1,2]. Gut microbiota modulates disease severity in various kidney diseases, including chronic kidney disease (CKD), diabetic kidney disease, glomerular nephritis, and acute kidney injury [3,4,5,6,7]. Microbiota is also involved in the CKD-related systemic complications such as cardiovascular diseases, uremic sarcopenia, and mineral bone disorders [8,9,10]. The effects along the gut-kidney axis mainly involve metabolic and immune pathways [1,11,12]. The microbiota presents both positive and negative metabolic effects, since it is responsible for producing beneficial metabolites in kidney diseases such as short-chain fatty acids [13,14] and harmful metabolites such as uremic toxins [15,16]. Regarding the immune response, microbiota plays key roles in the maturation and induction of immune cells such as regulatory T cells and Th17 cells [17,18,19].

Owing to the various roles played by the microbiota, its depletion either by germ-free conditions or by treatment with antibiotics can influence every kidney disease differently. For instance, studies have shown that antibiotic treatment attenuates kidney damage and renal inflammation via the suppression of inflammatory responses in ischemia-reperfusion damage [20], congenital kidney disease [21], and autoimmune glomerular nephritis [22,23]. In the autoimmune glomerular nephritis model, germ-free conditions also mitigated kidney damage by suppressing the migration of inflammatory cells such as Th17, to the kidney [22]. In contrast, germ-free mice show more severe renal dysfunction in ischemia-reperfusion damage [24]. Therefore, the extent of the influence of microbiota on kidney diseases appears to vary depending on the model used and methods of microbial depletion (germ-free condition or antibiotic treatment).

Adenine-induced kidney disease in mice is frequently used as an experimental animal model of CKD [4,9,25,26]. Mechanistically, orally administered adenine (a purine base) is converted into an insoluble nephrotoxic byproduct 2,8-dihydroxyadenine (2,8-DHA) that forms tubulointerstitial crystals and causes renal inflammation and fibrosis [27]. We have previously reported that germ-free (GF) mice showed more severe adenine-induced kidney damage than specific-pathogen-free (SPF) mice with normal microbiota [15]; however, the underlying mechanism for this has not yet been elucidated.

Additionally, the gut microbiota is also involved in the purine metabolism in the host. Microbiota plays a role in the symbiotic recycling of purine metabolites [28]. Gut microbiota produces and releases large quantities of purines, thus making them available to the intestinal mucosa [29]. Several microbial strains have a nucleosidase activity and degrade purine nucleosides in the intestines [30]. Thus, the alteration of the host purine metabolism by loss of microbiota may contribute to the more severe adenine-induced kidney damage in the GF mice. In the present study, to clarify this causative mechanism, we examined: (i) the effect of microbiota on the host purine metabolism; and (ii) the influence of loss of microbiota on the kidney damage and renal immune response in the adenine-induced CKD mice.

## 2. Results

### 2.1. The Germ-Free Condition Increased the Expression of Purine Metabolizing Enzymes

When excessive adenine is administered, it is converted into 2,8-DHA by xanthine dehydrogenase (XDH), which also converts hypoxanthine into uric acid (Figure 1a,b). The final product of the purine metabolism in rodents is allantoin (Figure 1b). To examine the effect of microbiota on the host purine metabolism, we evaluated the expression levels of purine metabolizing enzymes and allantoin, between SPF-normal diet (SPF + ND) and GF-normal diet (GF + ND) mice. XDH expression in the liver and kidney was significantly higher in GF mice than that in SPF mice (Figure 1c), and GF mice showed higher levels of urinary excretion of allantoin (Figure 1d). This suggested the activation of XDH pathway in GF mice. In addition, the livers of GF mice showed significantly higher expression of other purine metabolizing enzymes, including phosphoribosyl pyrophosphate synthetase 1 (PRPS1), purine nucleoside phosphorylase 1 (PNP1), adenine phosphoribosyl transferase (APRT), and hypoxanthine phosphoribosyl transferase (HPRT) (Figure 1e). These findings suggest that the purine metabolism pathway was enhanced in GF mice compared with SPF mice.

### 2.2. Fecal Purine Metabolites Were Altered between GF and SPF Conditions

To further examine the effect of gut microbiota on purine metabolism, we evaluated purine metabolites in the feces of the GF + ND and SPF + ND mice (Figure 2). Significant differences were found in the levels of each fecal purine metabolite between the GF and SPF conditions. The levels of guanosine, inosine, xanthine, and urate were significantly higher in the feces of GF + ND mice than those in the feces of SPF + ND mice; in contrast, levels of guanosine monophosphate (GMP), adenosine monophosphate (AMP), guanine, adenosine, adenine, and hypoxanthine were significantly lower in the GF + ND mice compared with those in the SPF + ND mice. From these findings, it is possible to speculate that gut microbiota affected purine metabolism in the intestine, which in turn would influence purine metabolism by the host, thereby promoting purine metabolism in GF mice. In addition, we measured the purine metabolites in the plasma and urine of the GF + ND and SPF + ND mice (Appendix A). We found that plasma levels of inosine, guanosine, xanthine, and hypoxanthine were significantly lowered in the GF + ND mice. These findings also suggest that the GF condition influenced purine metabolism in the host.

### 2.3. Germ-Free Conditions Exacerbated Adenine-Induced Kidney Damage in Mice

To evaluate the effect of GF condition on kidney injury induced by adenine, we compared the renal conditions of four groups of mice: SPF-normal diet (SPF + ND), GF-normal diet (GF + ND), SPF-adenine diet (SPF + AD), and GF-adenine diet (GF + AD), using the samples and data in our previous study (Figure 3a). Although adenine feeding induced kidney damage both in the SPF + AD and GF + AD mice, the blood urea nitrogen and creatinine levels were significantly higher in the GF + AD than in the SPF + AD group, as reported previously [15] (Figure 3b). Histologically, the renal tubular damage and macrophage infiltration shown by Masson Trichrome and F4/80 staining, respectively, were also more severe in GF + AD than in SPF + AD mice (Figure 3c). Furthermore, the expression levels of fibrosis-related genes (*Col1a1*, *Tgfb1*, and *Fn1*) were higher in the kidneys of GF + AD mice when compared with SPF + AD mice. These findings indicated that adenine-induced kidney damage was more severe in GF mice than in SPF mice.

### 2.4. Antibiotic Treatment Did Not Exacerbate Adenine-Induced Kidney Damage

Next, we examined whether depletion of microbiota by antibiotic treatment replicates the effects of GF condition on the host purine metabolism and adenine-induced kidney damage (Figure 4a,c). However, the oral antibiotic treatment for five weeks did not result in significant changes in XDH expression in both the liver and kidney (Figure 4b). On comparing the renal conditions of the four mice groups (normal diet, ND; adenine diet, AD; normal diet with antibiotic treatment, ND + Abx; and adenine diet with antibiotic treatment, AD + Abx), the tubular damage, fibrosis, and macrophage infiltration in the kidney was less severe in the AD + Abx group than in the AD group (Figure 4d,e), although blood urea nitrogen and creatinine levels were not significantly altered between the AD and AD + Abx mice (Figure 4f). In addition, the expression levels of fibrosis-related genes (*Tgfb1*, *Fn1*, and *Col1a1*) and inflammation-related genes (*Emr1*, *Il1a*, and *Tnf*) were lower in the kidneys of AD + Abx mice than those in AD mice without antibiotic treatment, suggesting that the antibiotic treatment could ameliorate renal fibrosis and inflammation in this model (Figure 4g). Therefore, unlike the germ-free condition, the antibiotic treatment did not influence the expression of the purine-metabolizing enzyme or exacerbate adenine-induced kidney damage.

### 2.5. Differences in Renal Inflammatory Responses between GF and SPF Mice

Next, to evaluate the contribution of different inflammatory responses to the exacerbated adenine-induced kidney damage in GF mice, we evaluated the expression levels of immune-related genes in four groups of mice: SPF + ND, GF + ND, SPF + AD, and GF + AD. A quantitative PCR array analysis revealed the comprehensive transcriptional profile targeting immune-related genes in the kidneys (Figure 5a). When comparing the expression profiles of the GF + AD and SPF + AD mice, the greatest difference was found in *Il17a*, which encodes interleukin (IL)-17A (Figure 5b). Individual PCR analysis confirmed that the expression level of *Il17a* was significantly increased in SPF + AD compared with that in SPF + ND mice; however, *Il17a* expression was undetectable in the GF + ND mice and was significantly lower in GF + AD than that in SPF + AD mice (Figure 5c). Similarly, SPF + AD mice showed a tendency of increased plasma IL-17 levels when compared with SPF + ND mice, and GF + AD mice presented lower plasma IL-17 levels than SPF + AD mice (Figure 5d). The transcription levels of IL-17F (a subfamily of IL-17), IL-23 (an inducing factor of IL-17A), and Foxp3 (a marker for regulatory T cells) were altered in the kidneys of both adenine-induced kidney injury groups; the levels were comparable between SPF and GF groups (Figure 5e).

### 2.6. Deficiency of IL-17A Did Not Influence Adenine-Induced Kidney Damage

To evaluate the role of IL-17A in adenine-induced kidney damage, we compared *Il17a*-/- and wild-type mice (Figure 6a). When mice were fed the adenine-rich diet, the degree of kidney dysfunction (measured by the elevation of blood urea nitrogen and creatinine) was not significantly different between *Il17a*-/- and wild-type mice (Figure 6b). Renal fibrosis and macrophage infiltration were also comparable between the two groups (Figure 6c). In addition, IL-17A deficiency did not significantly affect the expression levels of XDH and fibrosis-related genes in the kidney (Figure 6d). These results suggest that although IL-17A was depleted in the kidneys of GF mice, it did not have a significant effect on the adenine-induced kidney damage.

## 3. Discussion

The present study demonstrated that the GF mice displayed increased expression of purine metabolizing enzymes, including XDH, which could promote the conversion of the administered adenine into the nephrotoxic 2,8-DHA, and result in exacerbated adenine-induced kidney damage. The renal immune response was also significantly altered by the GF condition, wherein IL-17A expression was the most affected in GF + AD mice; however, the deletion of this gene did not significantly affect the disease condition in our adenine-induced kidney damage model. These observations suggest that the microbiota modulates host purine metabolism, which would be involved in the exacerbation of adenine-induced kidney damage in GF mice (Figure 7).

The conversion of administered adenine into 2,8-DHA by XDH is the key step in the induction of kidney injury by adenine feeding, which was demonstrated by the amelioration of kidney injury upon treatment with XDH inhibitors [31,32] and by the less severe kidney damage in mice with low expression level of XDH [33]. Thus, the differences in XDH expression levels observed in our study could modulate the severity of adenine-induced kidney damage and an increased XDH expression level could plausibly be the cause of exacerbated adenine-induced kidney damage in GF mice. According to our data, in addition to XDH, other purine metabolizing enzymes presented higher expression levels in GF mice than in SPF mice. This effect of the loss of microbiota on the expression of metabolizing enzymes in the host has also been reported by other studies. For example, most of the major cytochrome P450 isozymes, which are xenobiotic-metabolizing enzymes, showed lower expression levels in GF mice when compared with SPF mice [34].

The gut microbiota plays a role in the symbiotic recycling of purine metabolites [28]. Several microbial strains have a nucleosidase activity and thus can degrade purine nucleosides into purine bases in the intestines [30]. Indeed, our findings indicated that purine nucleosides such as guanosine and inosine were present at significantly lower levels in the feces of SPF mice than in that of GF mice; in addition, purine bases such as guanine and hypoxanthine were significantly higher in the feces of SPF mice (Figure 2). This indicates that the purine nucleosides were in fact converted into purine bases by the microbiota. In GF mice, the loss of purine metabolism performed by the microbiota may have enhanced purine metabolism in the host, which could potentially be reflected by the increased expression of purine-metabolizing enzymes in GF mice (Figure 7). However, future research needs to be performed to elucidate the detailed pathways and signals connecting the purine metabolism by the microbiota and host.

IL-17 is a cytokine having ambivalent proinflammatory and anti-inflammatory functions that are likely to be dependent on the tissue and environment [35,36]. Commensal microbiota is required for the maturation and differentiation of IL-17-producing lymphocytes such as Th17 cells [18,19]. In the present study, we demonstrated that IL-17A was the most altered cytokine in injured kidneys in the GF condition, but its depletion did not show significant effects on adenine-induced kidney damage. Although IL-17 has been reported to be involved in the pathogenesis of autoimmune glomerular nephritis [37], its signaling shows a protective effect on renal fibrosis in the obstructed kidney model mice [38]. In addition, another study showed that IL-17A was not involved in the progression of CKD in 5/6 nephrectomy model mice [39]. Therefore, the role of IL-17 appears to be different in each kidney disease and may not be extensive in the adenine-induced CKD model.

Our results showed the different effects of a GF condition and an antibiotic treatment on the adenine-induced CKD condition and XDH expression. Although both conditions involve depletion of the microbiota, the effects of the congenital depletion in GF mice are not necessarily the same as those of an antibiotic treatment [40]. The increased expression of purine metabolizing enzymes in GF mice was not replicated by the antibiotic treatment, suggesting that an exposure to microbiota in early life may be necessary for modulating host purine metabolism. This relationship seems to be linked to the developmental origins of health and disease (DOHaD) hypothesis [41], in which early-life exposure is connected to conditions displayed later in life. In addition, it has been reported that antibiotic treatment does not completely destroy the gut microbiota and causes dysbiosis [42]. Thus, the remaining antibiotic-resistant microbiota might perform purine metabolism in the intestine, which does not lead to the alteration of host purine metabolism in mice treated with antibiotics. Similar findings regarding susceptibility to renal injury have been reported in other kidney damage models. Antibiotic treatment ameliorated renal ischemia-reperfusion damage by dampening the renal inflammatory response [17]; in contrast, GF mice presented enhanced ischemia-reperfusion damage when compared with conventional mice [24]. Differences in immune response modulation with or without early exposure to microbial stimuli are considered as the cause for these opposing findings [24]. Therefore, when examining the effects of microbiota on the host, the differences between GF models and antibiotic treatments should be cautiously considered.

## 4. Conclusions

Our study demonstrated the importance of microbiota in the regulation of host purine metabolism. This could be responsible for modulating the disease condition in the kidney injury model, especially when the adenine/purine metabolism is involved. When examining the effects of certain interventions such as drugs and genetic modifications on adenine-induced kidney injuries, we need to consider the possible effects on the purine metabolism including the expression and activity of XDH.

## 5. Materials and Methods

### 5.1. Animal Experiments

All animal experiments were approved by the Animal Committee of Tohoku University, School of Medicine (approval No. 2019-BeA012 approved on 25 April 2019, No. 2019-BeA014 approved on 1 April 2019). Male IQI, C57BL/6Njc1, and C57BL/6JJcl mice were purchased from Clea Japan (Tokyo, Japan). *Il17a*-/- mice were previously described [43]. For Experiment 1, IQI GF mice were housed in vinyl isolators and provided with sterilized water and normal chow CE-2 pellets (Clea Japan) [44] *ad libitum*. IQI SPF mice were kept under SPF conditions and fed CE-2 pellets and tap water. At seven weeks of age, each group of mice was divided into normal and adenine diet groups and fed the normal CE-2 diet or a 0.2% adenine-containing CE-2 diet (Clea Japan, Tokyo, Japan) for five weeks. In the final two weeks of this period in the adenine diet group, a normal diet was fed for two days per week because continuance of the adenine diet would enfeeble the GF mice. These mice and samples of this experiment were the same set used in our previous study [15]. For Experiment 2, at eight weeks of age, C57BL/6Njc1 mice were divided into control and antibiotic groups. The mice were fed the normal CE-2 diet and tap water +/− a cocktail of antibiotics (ampicillin 0.5 g/L, neomycin 0.5 g/L, metronidazole 0.5 g/L, and vancomycin 0.25 g/L) continuously via water bottle for four weeks. To hide the flavor of antibiotics and to prevent mice dehydration, we supplemented control and antibiotic water with an artificial sweetener, Equal^®^ (5 g/L), as described previously [45]. For Experiment 3, at eight weeks of age, male C57BL/6Njc1 mice were divided in four groups: normal diet +/− antibiotics and adenine diet +/− antibiotics, followed by one week-of normal diet and antibiotics via water bottle for the antibiotics group. The mice were fed the normal CE-2 diet or the 0.2% adenine-containing diet and tap water supplemented by the artificial sweetener (5 g/L) +/− the cocktail of antibiotics continuously via water bottle for five weeks. At the end of each study, the mice were euthanized under isoflurane-induced anesthesia, and samples were collected. For Experiment 4, at eight weeks of age, male *Il17a*-/- mice and wild-type C57BL/6JJcl mice were fed the normal CE-2 diet or the 0.2% adenine-containing diet for six weeks.

### 5.2. Quantitative PCR

Total RNA extracted with RNeasy Mini Kit (Qiagen, Chatsworth, CA, USA) from the liver and kidneys was transcribed using ReverTra Ace qPCR RT Master Mix with gDNA Remover (FSQ30, Toyobo, Osaka, Japan). Quantitative PCR was performed using THUNDERBIRD Probe qPCR Mix (Toyobo) and StepOne plus (Thermo Fisher Scientific, Waltham, MA, USA) as described previously [4,46], and the Taqman probes used are listed in Appendix A. The values obtained were used in the ΔΔCt method of relative quantification with normalization to 18S gene expression.

### 5.3. Immune Array Analysis

The TaqMan Array Mouse Immune Panel (Thermo Fisher Scientific, Waltham, MA, USA) was used to determine the immune expression profile of the kidney.

### 5.4. Measurement of Allantoin and Purine Metabolites

Plasma, urinary, and fecal levels of fecal purine metabolites including allantoin were measured using capillary electrophoresis-time-of-flight mass spectrometry (CE-TOFMS) as described previously [15]. All CE-TOFMS experiments were performed using an Agilent CE capillary electrophoresis system (Agilent Technologies, Santa Clara, CA, USA), an Agilent G3250AA LC/MSD TOF system (Agilent Technologies), an Agilent 1100 series binary HPLC pump, a, G1603A Agilent CE-MS adapter, and a G1607A Agilent CE-ESI-MS sprayer kit.

### 5.5. Measurement of Renal Function Parameters

Blood urea nitrogen was assessed using a blood analyzer (i-STAT; Fuso Pharmaceutical Industries). Creatinine was measured using CE-TOFMS in Experiment 1 and a blood analyzer (i-STAT) in Experiment 2, 3, and 4.

### 5.6. Measurement of Plasma IL-17

Plasma IL-17 levels were measured using Mouse IL-17 Immunoassay (R&D Systems, Minneapolis, MN, USA).

### 5.7. Histology

Tissues were fixed in 10% neutral buffered formalin and embedded in paraffin. Kidney sections were stained with Masson’s trichrome, and Sirius red [4,47]. For immunohistochemistry of F4/80, antigen retrieval from deparaffinized sections was performed using Target Retrieval Solution S1700 (Dako, Carpinteria, CA, USA) in an autoclave at 120 °C for five min. An anti-F4/80 monoclonal antibody (1:100 dilution, MCA497, Bio-Rad, Hercules, CA, USA) was used as the primary antibody. For the quantitative analysis, the remaining tubules in the areas of the cortex were evaluated in Masson’s trichrome-stained sections, using ImageJ software (National Institutes of Health, Bethesda, MD, USA).

### 5.8. Statistical Analysis

Values are presented as mean ± standard error of the mean (S.E.M.). Statistical analyses were conducted using JMP 14.1 software (SAS Institute, Cary, NC, USA). The Shapiro–Wilk normality test was used to test the normality of data. One-way analysis of variance (ANOVA) with Dunnett’s or Tukey post hoc test was used to compare more than two group means. Statistical comparisons between two groups were performed using two-tailed Student’s *t*-test. Results were considered significant at *p* values < 0.05. The heat map was constructed using JMP 14.1.

## Figures and Tables

**Figure 1 toxins-12-00547-f001:**
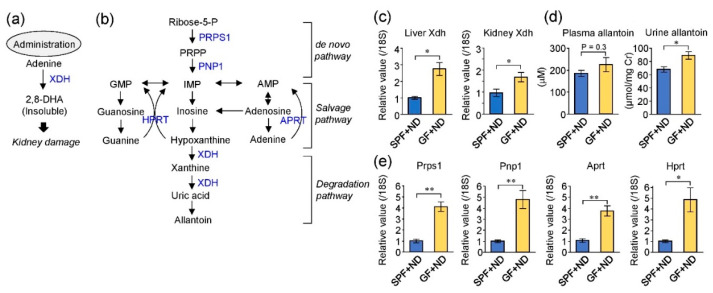
Germ-free (GF) conditions increased the expression of purine metabolizing enzymes. (**a**) Pathway of kidney damage by the administration of adenine. 2,8-DHA, 2,8-dihydroxyadenine; XDH, xanthine dehydrogenase. (**b**) Metabolic pathway of purines. PRPS1, phosphoribosyl pyrophosphate synthetase 1; PNP1, purine nucleoside phosphorylase 1; APRT1, adenine phosphoribosyl transferase; HPRT1, hypoxanthine phosphoribosyl transferase. (**c**) Expression of XDH mRNA in the liver and kidney. Specific-pathogen-free (SPF) + ND, SPF mice with normal diet; GF + ND, germ-free mice with normal diet. (**d**) Plasma and urine allantoin levels. Urinary concentrations were corrected for urinary creatinine (μmol/mg urinary creatinine). (**e**) mRNA levels of genes corresponding to purine metabolizing enzymes in the liver. * *p* < 0.05, ** *p* < 0.01 compared between indicated groups (*t*-test). *n* = 4 in SPF + ND and GF + ND groups.

**Figure 2 toxins-12-00547-f002:**
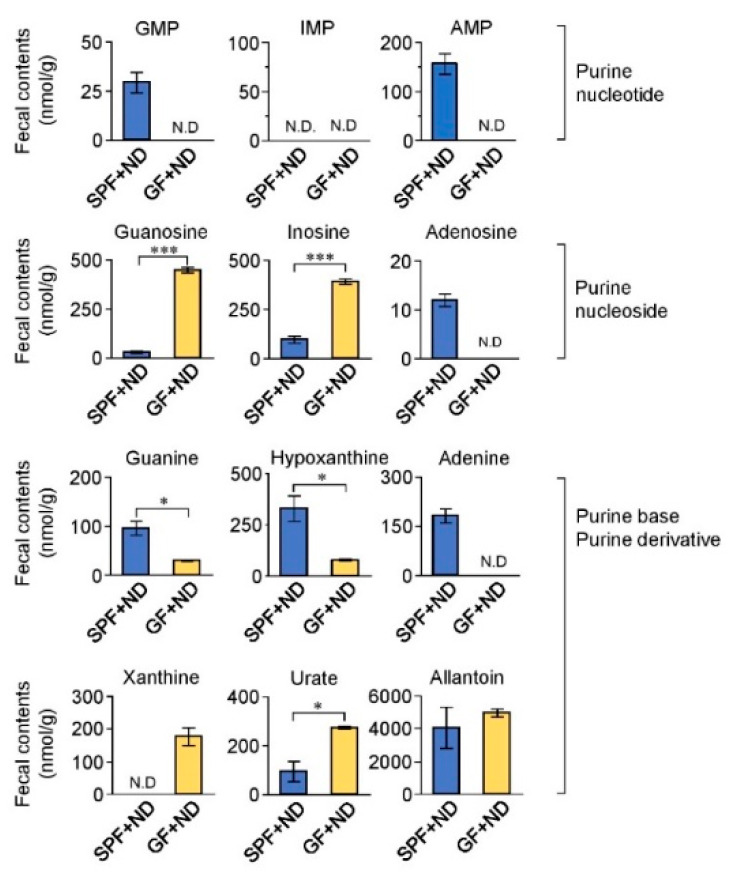
Fecal nucleotide metabolites in GF and SPF mice. Fecal samples obtained from GF + ND and SPF + ND mice were measured. Fecal concentrations were corrected for fecal weight (nmol/g). * *p* < 0.05, *** *p* < 0.001 compared between indicated groups (*t*-test). N.D, not detectable.

**Figure 3 toxins-12-00547-f003:**
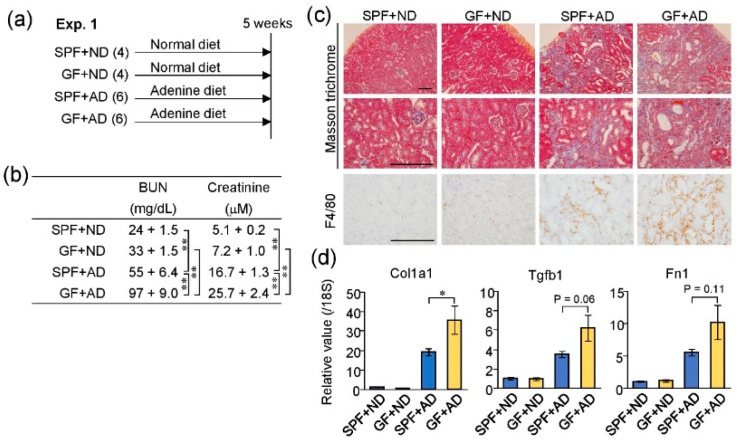
Germ-free (GF) conditions exacerbated adenine-induced kidney damage. (**a**) Experimental course of adenine-induced kidney injury in mice raised under SPF or GF conditions. The numbers of used mice are shown in brackets. ND, normal diet; AD, adenine diet. (**b**) Levels of blood urea nitrogen (BUN) and plasma creatinine (mean ± S.E.M). This result is quoted from our previous study [15]. (**c**) Histological images of kidney sections with Masson’s trichrome and F4/80 immunohistochemistry. Scale bar, 200 μm. (**d**) mRNA levels of fibrosis-related genes in the kidney. * *p* < 0.05, ** *p* < 0.01 compared between indicated groups (ANOVA).

**Figure 4 toxins-12-00547-f004:**
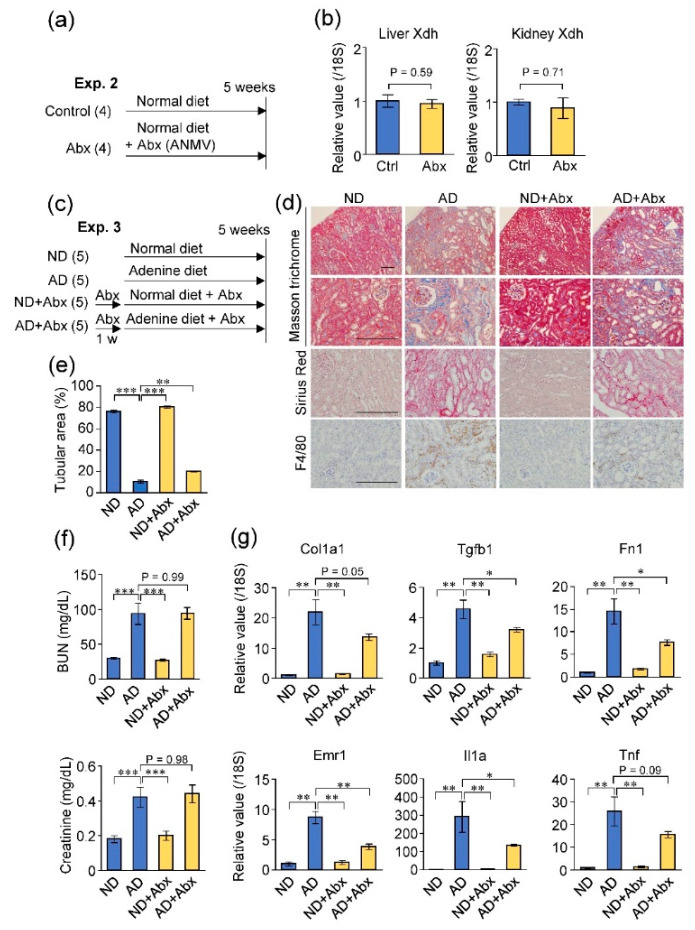
Antibiotic treatment did not increase XDH expression or exacerbate the adenine-induced kidney damage. (**a**) Experimental course of normal mice treated with or without antibiotics (Abx). ANMV; ampicillin, neomycin, metronidazole, and vancomycin. (**b**) Expression levels of XDH mRNA in the liver and kidney of control (Ctrl) and antibiotic-treated mice. (**c**) Experimental course of adenine-induced kidney injury in mice treated with or without antibiotics. (**d**) Histological analysis of the kidney sections with Masson’s trichrome, Sirius red and F4/80 stain. Scale bar, 200 μm. (**e**) Quantitative analysis of the remaining tubular area in the cortex indicated by Masson’s trichrome staining. (**f**) Blood urea nitrogen (BUN) and creatinine concentration. (**g**) Expression levels of fibrosis- (*Tgfb1*, *Fn1*, and *Col1a1*) and inflammation-related (*Emr1*, *Il1a*, and *Tnf*) genes in the mice kidneys. * *p* < 0.05, ** *p* < 0.01, *** *p* < 0.001 compared between indicated groups (ANOVA).

**Figure 5 toxins-12-00547-f005:**
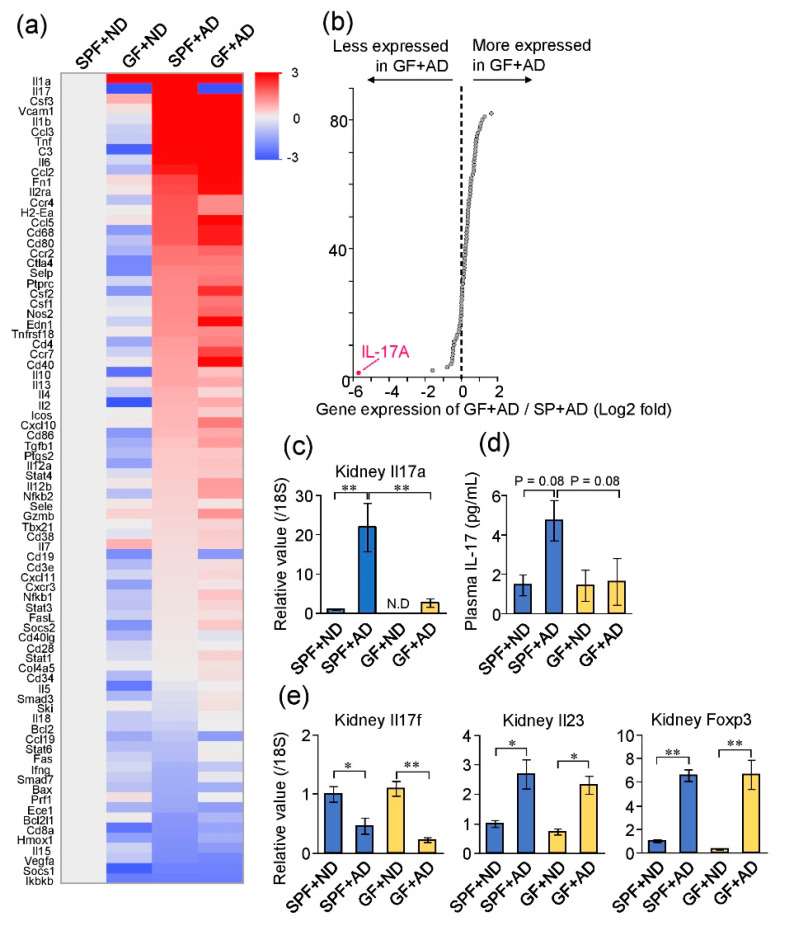
Profiling of immuno-related genes in the adenine-induced damaged kidney of mice under GF and SPF conditions. (**a**) Transcriptional profiling of immuno-related genes in the kidneys of the four mice groups: SPF + ND, SPF + AD, GF + ND, and GF + AD (in the experiment described in Figure 3a). (**b**) Scatter plot comparing expression levels of SPF + AD and GF + AD groups. (**c**) Expression level of *Il17a* mRNA in the kidney. (**d**) Plasma IL-17 levels. (**e**) Expression levels of *Il17f*, *Il23*, and *Foxp3* mRNA in the kidney. * *p* < 0.05, ** *p* < 0.01 compared between indicated groups (ANOVA).

**Figure 6 toxins-12-00547-f006:**
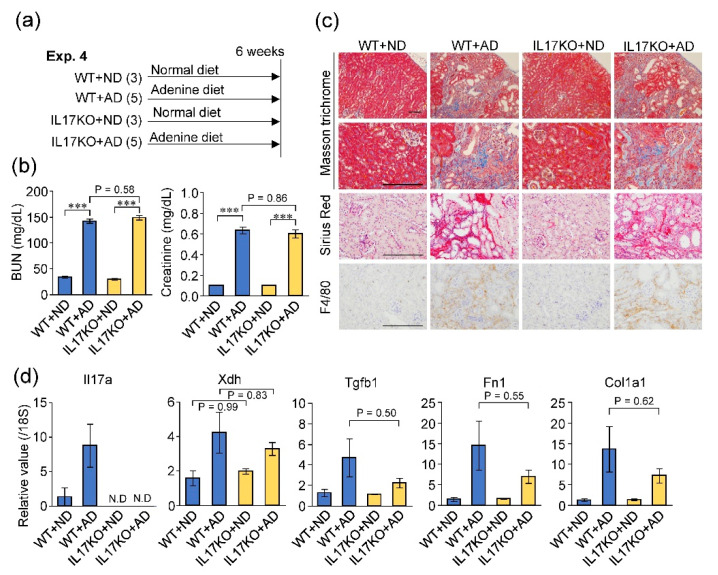
Deficiency of IL-17A did not affect adenine-induced kidney damage. (**a**) Experimental course of wild-type (WT) and IL-17A knockout (IL17 KO) mice treated with or without adenine. (**b**) Blood urea nitrogen (BUN) and creatinine levels. (**c**) Histological analysis of the kidney sections using Masson’s trichrome, Sirius red and F4/80 stain. Scale bar, 200 μm. (**d**) mRNA expression levels of IL-17A, XDH, and fibrosis-related genes in the kidney. *** *p* < 0.001 compared between indicated groups (ANOVA).

**Figure 7 toxins-12-00547-f007:**
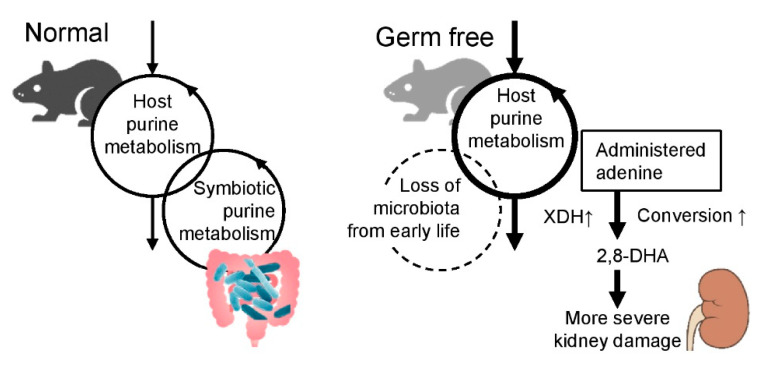
Schematic representation of purine metabolism in normal and germ-free mice.

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
