# Peer review of "Germ-Free Conditions Modulate Host Purine Metabolism, Exacerbating Adenine-Induced Kidney Damage"

_toxins, 2020, doi:10.3390/toxins12090547_

Round 1

Reviewer 1 Report

The study is interesting and the results are of good quality.
The author lacks to discuss the lack of nucleotide/nucleoside profiling in blood and urine (qualitative research and quantitative assay) that can link genotype and final phenotype of enzymes.

The paragraph of stat analysis is a little light: before proposing a stat test, the underlying hypotheses must be verified

Author Response

Reviewer 1

#1. The author lacks to discuss the lack of nucleotide/nucleoside profiling in blood and urine (qualitative research and quantitative assay) that can link genotype and final phenotype of enzymes.

Response to comment#1. We appreciate the reviewer’s valuable comments. We added the results of nucleotide/nucleoside profiling in blood and urine of SPF and GF mice in supplementary figure 1 and 2. As a result, plasma levels of inosine, guanosine, xanthine, and hypoxanthine were significantly lowered in the GF mice than SPF mice. These findings also suggest the GF condition influenced the purine metabolism in the host body. About these results, we added the description as follows.

Line 104-108: “In addition, we measured the purine metabolites in the plasma and urine of the GF+ND and SPF+ND mice (Supplementary figure 1 and 2). We found that plasma levels of inosine, guanosine, xanthine, and hypoxanthine were significantly lowered in the GF+ND mice. These findings also suggest that the GF condition influenced purine metabolism in the host.”

Line 304-307: Plasma, urinary, and fecal levels of fecal purine metabolites including allantoin were measured using capillary electrophoresis-time-of-flight mass spectrometry (CE-TOFMS) as described previously.

#2. The paragraph of stat analysis is a little light: before proposing a stat test, the underlying hypotheses must be verified.

Response to comment #2. We appreciate the reviewer's suggestion. Following this, we have added the sentence about the statistical hypothesis testing in the method section as follows.

Line 326-332 “Statistical comparisons between two groups were performed using two-tailed Student’s t-test. One-way analysis of variance (ANOVA) with Dunnett’s or Tukey post hoc test was used to compare more than two group means. The Shapiro–Wilk normality test was used to test the normality of data. Results were considered significant at P values < 0.05.”

Reviewer 2 Report

The current study aimed to define the role of the gut microbiota in purine metabolism and its effects on the adenine induced kidney toxicities. Overall the study describes multiple animal experiments with interesting finding but the manuscript was hard to read. Below are few suggestions that can help improve the manuscript.

In the introduction authors did not mention whether microbiota has been shown to degrade or produce adenine and other purine metabolites. It is mentioned in the discussion but important  to add the information in the introduction. It is unclear what is the hypothesis of the authors regarding the protective effect of the microbiota with regards to adenine toxicity and how this is linked to overall purine metabolism.

I suggest the authors describe the two distinct questions they are exploring in the introduction because they were not clear. I also think rearranging the sections to answer each question will make the manuscript easier to read.  

1) what is the effect of microbiome on kidney injury induced by adenine? For this experiment diet is high adenine. The results are in section 2.1, 2.4. 2.5 and 2.6.

 2) what is the effect of the microbiome on purine metabolism in general? For this experiment diet was not high in adenine and results are in section 2.2 and 2.3

In each results section, adding a sentence or 2 explaining the rationale and hypothesis for each experiment will also be helpful.

Figure 1 panel b. are the numbers  mean with sd? Is BUN/ Cr comparable between SPF and GF or significantly different?

Do you have information on the purine content of normal diet? I assume it is constant between experiments.

Author Response

Reviewer 2

The current study aimed to define the role of the gut microbiota in purine metabolism and its effects on the adenine induced kidney toxicities. Overall the study describes multiple animal experiments with interesting finding but the manuscript was hard to read. Below are few suggestions that can help improve the manuscript.

Response to comment:

We appreciate the reviewer’s instructive suggestion. According to the advices and comments, we revised the manuscript as follows.

#1: In the introduction authors did not mention whether microbiota has been shown to degrade or produce adenine and other purine metabolites. It is mentioned in the discussion but important to add the information in the introduction. It is unclear what is the hypothesis of the authors regarding the protective effect of the microbiota with regards to adenine toxicity and how this is linked to overall purine metabolism.

Response to #1. We appreciate the reviewer’s comment. According to your suggestion, we added description about the known role of microbiota in the purine metabolism, and clearly described the hypothesis regarding the effect of the microbiota on adenine toxicity in introduction.

Line 59-67: “Additionally, the gut microbiota is also involved in the purine metabolism in the host. Microbiota plays a role in the symbiotic recycling of purine metabolites [28]. Gut microbiota produces and releases large quantities of purines, thus making them available to the intestinal mucosa [29]. Several microbial strains have a nucleosidase activity, and degrade purine nucleosides in the intestines [30]. Thus, the alteration of the host purine metabolism by loss of microbiota may contribute to the more severe adenine-induced kidney damage in the GF mice. In the present study, to clarify this causative mechanism, we examined i) the effect of microbiota on the host purine metabolism, and ii) the influence of loss of microbiota on the kidney damage and renal immune response in the adenine-induced CKD mice”

#2. I suggest the authors describe the two distinct questions they are exploring in the introduction because they were not clear. I also think rearranging the sections to answer each question will make the manuscript easier to read. 

1) what is the effect of microbiome on kidney injury induced by adenine? For this experiment diet is high adenine. The results are in section 2.1, 2.4. 2.5 and 2.6.

 2) what is the effect of the microbiome on purine metabolism in general? For this experiment diet was not high in adenine and results are in section 2.2 and 2.3

Response to #2. According to the reviewer’s suggestion, we rearranged the section order as follows.

1) The effect of the microbiota on purine metabolism (section 2.1, 2.2).

2) The effect of microbiota on adenine-induced kidney injury (section 2.3, 2.4).

3) The effect of microbiota on immune-response in adenine-induced kidney injury (section 2.5, 2.6).

#3. In each results section, adding a sentence or explaining the rationale and hypothesis for each experiment will also be helpful.

Response to #3. We appreciate the valuable advice. We added the explanation about the purpose and rationale in each result section to make the manuscript easier to read as follows.

Line 72-74: To examine the effect of microbiota on the host purine metabolism, we evaluated the expression levels of purine metabolizing enzymes and allantoin.

Line 115: To evaluate the effect of GF condition on kidney injury induced by adenine, ---

Line 136: Next, we examined whether depletion of microbiota by antibiotic treatment replicates the effects of GF condition on the host purine metabolism and adenine-induced kidney damage

Line 163-165: Next, to evaluate the contribution of different inflammatory responses to the exacerbated adenine-induced kidney damage in GF mice, we evaluated the expression levels of immune-related genes in four groups of mice

#4. Figure 1 panel b. are the numbers mean with sd? Is BUN/ Cr comparable between SPF and GF or significantly different?

Response to #4. The values in the Figure 1b is mean +- SEM. We added the explanation in the legend of Figure 3. BUN and Cr levels were not significantly different between SPF+ND and GF+ND. We added the P value in the panel (P=0.81 in BUN, P=0.87 in Cr).

#5 Do you have information on the purine content of normal diet? I assume it is constant between experiments.

Response to #5. Thank you for the comment. The manufacture makes the normal diet (CE-2) with a fixed composition without addition of adenine and other purines (44), indicating the purine content in the normal diet is constant. However, the adenine content in the normal diet is little because the measured adenine content level was ~0.7% in the CE-2 diet added 0.7% adenine (company data of Japan Clea). In the method section, we added the reference addressing the composition of the normal diet (Ref#44, line 275).